# Decongestion in Acute Heart Failure—Time to Rethink and Standardize Current Clinical Practice?

**DOI:** 10.3390/jcm13020311

**Published:** 2024-01-05

**Authors:** Valentin Bilgeri, Philipp Spitaler, Christian Puelacher, Moritz Messner, Agne Adukauskaite, Fabian Barbieri, Axel Bauer, Thomas Senoner, Wolfgang Dichtl

**Affiliations:** 1Department of Internal Medicine III, Medical University of Innsbruck, 6020 Innsbruck, Austria; valentin.bilgeri@i-med.ac.at (V.B.); philipp.spitaler@i-med.ac.at (P.S.); christian.puelacher@i-med.ac.at (C.P.); moritz.messner@tirol-kliniken.at (M.M.); agne.adukauskaite@tirol-kliniken.at (A.A.); axel.bauer@i-med.ac.at (A.B.); 2Deutsches Herzzentrum der Charité, Hindenburgdamm 30, 12203 Berlin, Germany; fabian.barbieri@hotmail.com; 3Department of Anesthesiology, Medical University of Innsbruck, 6020 Innsbruck, Austria; thomas.senoner@i-med.ac.at

**Keywords:** natriuresis, spot urinary sodium, desalination, combination diuretic therapy

## Abstract

Most episodes of acute heart failure (AHF) are characterized by increasing signs and symptoms of congestion, manifested by edema, pleura effusion and/or ascites. Immediately and repeatedly administered intravenous (IV) loop diuretics currently represent the mainstay of initial therapy aiming to achieve adequate diuresis/natriuresis and euvolemia. Despite these efforts, a significant proportion of patients have residual congestion at discharge, which is associated with a poor prognosis. Therefore, a standardized approach is needed. The door to diuretic time should not exceed 60 min. As a general rule, the starting IV dose is 20–40 mg furosemide equivalents in loop diuretic naïve patients or double the preexisting oral home dose to be administered via IV. Monitoring responses within the following first hours are key issues. (1) After 2 h, spot urinary sodium should be ≥50–70 mmol/L. (2) After 6 h, the urine output should be ≥100–150 mL/hour. If these target measures are not reached, the guidelines currently recommend a doubling of the original dose to a maximum of 400–600 mg furosemide per day and in patients with severely impaired kidney function up to 1000 mg per day. Continuous infusion of loop diuretics offers no benefit over intermittent boluses (DOSE trial). Emerging evidence by recent randomized trials (ADVOR, CLOROTIC) supports the concept of an early combination diuretic therapy, by adding either acetazolamide (500 mg IV once daily) or hydrochlorothiazide. Acetazolamide is particularly useful in the presence of a baseline bicarbonate level of ≥27 mmol/L and remains effective in the presence of preexisting/worsening renal dysfunction but should be used only in the first three days to prevent severe metabolic disturbances. Patients should not leave the hospital when they are still congested and/or before optimized long-term guideline-directed medical therapy has been initiated. Special attention should be paid to AHF patients during the vulnerable post-discharge period, with an early follow-up visit focusing on up-titrate treatments of recommended doses within 2 weeks (STRONG-HF).

## 1. Introduction

Heart failure (HF) is a continuously growing global health care problem, affecting approximately 2–3% of the adult population, with a strong increase in the elderly and very elderly. It is a progressive, malignant and complex pathologic entity, with overall annular mortality rates reaching 10%. Although effective treatment options have improved outcomes, the prognosis is still unacceptably dismal, with a mean 5-year mortality rate around 50% [1]. Patients hospitalized due to acute heart failure (AHF) are at even higher risk. This holds true in particular for patients with persistent signs of residual congestion at discharge, with mortality in patients with hypervolemia and hypoperfusion reaching 50% in one year [2]. Despite best efforts, residual congestion at discharge is common and is present in 31–48% of patients [2,3]. Worsening renal function (WRF) during AHF, defined by an increase in serum creatinine of ≥0.3 mg/dl from values at admission, is in contrast no independent risk marker for a dismal outcome, but the combination of ongoing congestion and WRF seems to have the worst prognosis [4].

HF is a clinical syndrome with classical symptoms such as dyspnea, fatigue and/or orthopnea. In typical cases of acute cardiac decompensation, congestion develops due to fluid overload manifesting clinically in peripheral edema, pleural effusion, pulmonary crackles, ascites, hepatomegaly, and/or elevated jugular venous/intraabdominal pressures. Diagnosis and judgement of congestion can be challenging and should therefore be supported by imaging (chest X-ray, ultrasound of the pleura, lungs, inferior vena cava, ascites) and biomarkers (hematocrit, serum creatinine, natriuretic peptides plasma levels) [5,6,7,8].

A diminished stroke volume as a consequence of left or right ventricular dysfunction results in a reduced cardiac output and increased central venous pressure. Injuries to the myocardium, leading to impaired relaxation and/or contractility and the reduced ability of the heart to eject blood, are most often caused by coronary artery disease, arterial hypertension and/or diabetes mellitus. Concomitant common valvular pathologies (such as functional mitral and/or tricuspid regurgitation) and electrical diseases (such as atrial fibrillation and/or left bundle branch block) often add to the vicious cycle, finally resulting in AHF events. Prior non-adherence to neurohumoral drug therapy and/or infections are common triggers for such clinical decompensations and are the main cause of unplanned hospitalizations in the elderly (>65 years of age) in high-income countries.

Compensatory mechanisms counteracting the reduced cardiac output represent the baroreceptor reflex, which increases the firing rate, thus activating sympathetic adrenergic nerves to the heart and vasculature. Other mechanisms are humoral changes, such as increased renin release from the kidneys, which leads to the formation of angiotensin II and aldosterone. The release of vasopressin (antidiuretic hormone—ADH) from the posterior pituitary gland is also stimulated. Due to sympathetic stimulation of the adrenal glands, circulating catecholamines (epinephrin and norepinephrine) are increasingly released in the context of HF. This sympathetic stimulation results in vasoconstriction in both arterial and venous vessels, in order to maintain an adequate arterial pressure and venous tone [9].

Congestion is defined by extracellular fluid accumulation leading to increased cardiac filling pressures [10]. Poor cardiac function has been recognized to be a prerequisite for developing congestion. However, several other mechanisms contribute to increased fluid retention in HF patients. The kidneys react in the early phase of myocardial dysfunction with increased sodium and water retention to counteract the perceived arterial underfilling and consequent reduction in effective circulating blood volume. Sodium is interstitially stored in HF patients bound to large glycosaminoglycan networks and this buffer function may collapse in the event of AHF, leading to increased local interstitial fluid accumulation [11].

Simultaneously, baroreceptor activation and neurohumoral stimulation ensue, further promoting renal sodium and water retention. A more gradual accumulation of interstitial compartment fluid with an associated increase in interstitial tissue pressure develops over time. Initial sympathetic-driven vasoconstriction maintains organ perfusion pressure and also contributes to maintaining effective tissue perfusion dynamics [12,13]. Importantly, changes in total blood volume are not equally distributed between arterial and venous systems, as the venous system contains up to 70% of the total blood volume, mostly in high capacitance splanchnic veins [14]. Changes in the capacitance of splanchnic veins may also contribute to the development of congestion. Sympathetic activation of these veins rapidly increases effective circulatory volume by translocating the splanchnic venous reservoir to the effective circulatory volume (auto-transfusion), thus augmenting cardiac preload and maintaining output [15]. Ultimately, both changes in total blood volume (volume overload) and venous capacitance (volume redistribution) can lead to AHF.

## 2. Application of Diuretics, Diuretic Resistance and Early Monitoring of Natriuresis

The first goal in AHF is to break the vicious cycle of hypervolemia with diuretics. When administering diuretics, various factors should be taken into consideration. First of all, the timing of medication administration upon arrival at the clinic is crucial. The door to diuretic time, defined by the duration from patient arrival at the emergency department (ED) to the first intravenous (IV) diuretic injection (usually a loop diuretic), has traditionally been considered the first critical parameter. A **door to diuretic time ≤ 60 min** is associated with a lower in-hospital mortality rate in some [16] but not all registries [17], potentially reflecting a shorter “time-to-diagnosis”. Despite these conflicting results, IV loop diuretics should be applied in patients with AHF and congestion presenting in the ED as soon as possible and regardless of perfusion status. In AHF, low blood pressure should not delay IV-diuretics, unless the patient is in cardiogenic shock or isolated RV failure is suspected. Patients with new-onset AHF or those with chronic HF but without chronic use of oral diuretics usually respond adequately to 20–40 milligrams of furosemide applied IV. If a patient receives chronic oral diuretic treatment at home, the IV dose of a loop diuretic should be at least equivalent or even doubled. The additional administration of acetazolamide (500 mg IV) or hydrochlorothiazide or related thiazide compounds may be superior, which leads to the new concept of **door to combination diuretics time**. A new treatment algorithm on decongestion in patients suffering from AHF is presented in Figure 1. The different sites of action within the nephron of the various classes of diuretics is shown in Figure 2.

The **DOSE** trial [18] analyzing 308 patients with AHF found no difference whether loop diuretics are given as intermittent boluses (every 12 h) or as a continuous infusion. In a factorial design, a low dose (equivalent to the patient’s previous oral dose) versus a high dose (2.5 times the previous oral dose) were compared as well. Although the primary outcome was not changed by the different dosing regime, there was a nonsignificant trend toward a greater improvement in the patients´ global assessment in the high-dose group. Notably, the trial confirmed the prognostic relevance that patients should reach euvolemia within 72 h. This goal could be reached in only 15% of the study cohort, emphasizing the unmet need to improve current decongestion strategies and reaching a good natriuretic response in the majority of AHF patients.

AHF patients with previous oral diuretic therapy have a much lower response to IV loop diuretics than healthy individuals who urinate approximately three liters within 4–6 h after a single IV injection of 40 milligram of furosemide [19]. Indeed, diuretic dose requirements tend to increase over time in HF patients, a phenomenon referred to as diuretic resistance. The causes of diuretic resistance are multifactorial, ranging from decreased renal blood flow due to low cardiac output, sympathetic nervous system activation, nephron remodeling, renin-angiotensin-aldosterone system (RAAS) activation, to changes in pharmacokinetics and dynamics of diuretics. Furthermore, urine characteristics change dramatically between the first and the following days in AHF patients undergoing IV loop diuretic therapy (progressive decrease in urinary sodium and chloride), even if urine output remains the same relative to the diuretic dose (= diuretic efficiency) [20].

The **ROSE-AHF** trial [21] confirmed the importance of the natriuretic response to diuretic therapy as the key prognostic parameter in AHF patients, outperforming traditional estimates such as weight loss or urinary output/fluid balance. Sodium excretion < 2 g/day was defined as a poor natriuretic response and occurred in 28.5% of patients after 24 h. An intermediate natriuretic response was defined as 2 to 4 g/day, and an excellent natriuretic response was defined as >4 g/day. AHF patients with an impaired natriuretic response following IV diuretics (**urinary spot Na^+^ < 50–70 mmol/L measured after 2 h**) have poor long-term outcomes [19,22]. Due to this, such patients should be identified early to allow for timely treatment intensification.

The impact of such a **natriuresis-guided approach in the first few hours** after hospital admission is currently being assessed in multiple ongoing studies (Table 1).
(1)The **PUSH-AHF** study [23] randomized in an open-label design 310 patients with AHF and planned IV loop diuretic administration (mean age 74 years, 45% females) either to natriuresis-guided therapy (urine sodium measured at 2, 6, 12, 18, 24 and 36 h) or usual care. Diuretic treatment was intensified if urine sodium was <70 mmol/L. The co-primary outcome, natriuresis at 24 h, was significantly higher in the intervention group (409 versus 345 mmol; *p* = 0.0061). However, the co-primary outcome of all-cause mortality or HF rehospitalization within 180 days remained unchanged (31% in both treatment arms; *p* = 0.70).(2)The **ENACT-HF** trial [24] enrolled 401 patients at 29 centers in a prospective, open-label, nonrandomized design. During the first phase, enrolled patients were treated according to usual care at their center. During the second phase, all centers switched to a standardized protocol emphasizing adequate initial dosing (doubling the oral home dose up to 200 mg furosemide equivalent IV) and further doubling the initial IV dose after 6 h if urinary sodium was <50 mmol/L or urinary output was <100 mL/hour, then given twice daily. The primary outcome of total natriuresis after 1 day was met, with a mean natriuresis of 174 mmol in the standard of care arm and 282 in the intervention arm (+64% change; mean ratio: 1.64, 95% confidence interval 1.37–1.95, *p* < 0.001). Furthermore, a reduction in the hospital duration was observed in the intervention group compared to the standard of care group (5.8 vs. 7.0 days, mean ratio: 0.87, 95% confidence interval 0.77–0.99, *p* = 0.036). Of note, weight loss and congestion score were not significantly different between groups. The treatment was also deemed safe, as there were no differences in the markers of renal dysfunction, hypokalemia or hypotension between the two treatment arms.(3)The ongoing **ESCALATE** trial [25]—an open labeled 1:1 randomized trial—includes a total of 450 AHF patients with hypervolemia of at least 10 pounds of estimated excess volume to a natriuresis-guided approach versus usual care. As a urine sodium concentration alone does not ensure a net negative sodium balance if not interpretated in relation to the urine creatinine, this study uses a natriuretic response prediction equation to predict the expected cumulative sodium excretion [26].(4)The ongoing **DECONGEST** trial (NCT05411991) has a similar design (randomized, open-label) investigating whether serial assessment of urinary sodium after diuretic administration improves decongestion versus usual care in AHF patients. It recommends combination diuretic therapy by the upfront use of (1) acetazolamide 500 mg once daily in the absence of hypernatremia (>145 mmol/L) or metabolic acidosis (bicarbonate < 22 mmol/L); and by the upfront use of (2) oral chlorthalidone 50 mg once daily if eGFR < 30 mL/min/1.73 m^2^ or hypernatremia (>145 mmol/L). The protocol further recommends a full nephron blockade with IV acetazolamide 500 mg once daily, IV bumetanide 4 mg twice daily, oral chlorthalidone 100 mg once daily, and IV canrenoate 200 mg once daily if the urinary sodium concentration remains <80 mmol/L with persistent signs of congestion.

## 3. Early Up-Titration of Guideline-Directed Medical Therapy and Post-Discharge Management

Optimal treatment of patients suffering from AHF consists of both successful and complete decongestion followed by fast and closely monitored up-titration of guideline-directed medical therapy (GDMT). The recently published STRONG-HF trial [27] randomizing 1078 patients clearly proved the benefit of intensive post-discharge management aiming to reach an up-titration of neurohumoral medication to 100% of recommended doses within 2 weeks. The intervention arm included four outpatient visits in the first 2 months assessing clinical status as well as laboratory values, including N-terminal pro-B-type natriuretic peptide (NT-proBNP) concentrations. As compared to the standard of care, this high-intensity care significantly reduced the composite endpoint of HF rehospitalization or all-cause death within 180 days: from 23.3% to 15.2% (risk ratio 0.66, 95% confidence interval 0.50–0.86, *p* = 0.0021).

## 4. Pharmacokinetics, Pharmacodynamics and Related Clinical Issues of Different Diuretic Drugs

### 4.1. Loop Diuretics—Inhibitors of Na^+^-K^+^-2Cl^–^ Symporter

Widely used loop diuretics currently available for the treatment of HF include furosemide, torsemide and bumetanide (Table 2). Loop diuretics inhibit the Na^+^-K^+^-2Cl^–^ symporter at the apical surface of the thick ascending limb cells along the loop of Henle. Consequently, sodium, potassium and chloride are reabsorbed less, leading to an increased loss of these electrolytes. This draws water into the nephron and subsequently leads to increased diuresis. Furthermore, loop diuretics also inhibit Ca^2+^ and Mg^2+^ reabsorption in the thick ascending loop by abolishing the transepithelial potential difference that is the dominant driving force for reabsorption of these cations. Two varieties of Na^+^-K^+^-2Cl^–^ symporter exist. The “absorptive” symporter (called ENCC2, NKCC2, or BSC1) is expressed only in the kidney, is localized to the apical membrane and subapical intracellular vesicles of the thick ascending loop and is regulated by the cyclic AMP/PKA pathway. The “secretory” symporter (called ENCC3, NKCC1, or BSC2) is a “housekeeping” protein with a wider pattern of expression including epithelial cells, muscle cells, neurons and red blood cells. It is also expressed in the ear and may likely be responsible for the ototoxicity seen with loop diuretics. Furosemide has weak carbonic anhydrase–inhibiting activity and thus increases urinary excretion of HCO_3_^−^ and phosphate. All loop diuretics increase urinary K^+^ and titratable acid excretion. This effect is due in part to enhanced K^+^ secretion due to increased distal tubular fluid flow. Other mechanisms contributing to enhanced K^+^ and H^+^ excretion include flow-dependent enhancement of ion secretion by the collecting duct, nonosmotic vasopressin release, and activation of the RAAS axis (for review, see [28]).

**Furosemide versus torasemide**: Following discussions on whether torsemide offers advantages as compared to furosemide in terms of both efficacy and safety for HF patients, the randomized, unblinded **TRANSFORM-HF** trial did not identify significant differences concerning survival, all-cause hospitalization, total hospitalizations or quality of life after 12 months in 2859 hospitalized AHF patients ([29,30]; for meta-analysis, see [31]).

**Subcutaneous furosemide administration**: Recent data from two phase I studies indicate that subcutaneous patch infusor devices demonstrate a similar bioavailability to IV administration of furosemide [32]. In the future, this method of administration could potentially be performed at home, thereby helping to prevent and reduce hospitalizations of patients at high risk to develop AHF.

**Switch from IV to oral loop diuretic therapy**: After clinical stabilization of the patient hospitalized due to AHF, IV loop diuretics should be switched to oral therapy and—if possible—progressively decreased [5]. In chronic HF, only the lowest amount of loop diuretics should be given, and even a complete wean of could be considered on an individual basis during long-term follow-up [33].

### 4.2. Thiazide-Inhibitors of Na^+^-Cl^−^ Symport

This class includes benzothiadiazine derivatives (thiazide or thiazide-type diuretics) and drugs that are pharmacologically similar to thiazide diuretics but differ structurally (thiazide-like diuretics) (Table 3). Thiazide diuretics act in the distal convoluted tubule by blocking the Na^+^-Cl^−^ symporter (called ENCC1 or TSC), which is predominantly expressed in the kidney. The expression of the symporter is regulated by aldosterone. Thiazide diuretics increase Na^+^ and Cl^−^ excretion. However, they are only moderately efficacious (i.e., maximum excretion of the filtered Na^+^ load is only 5%, compared to loop diuretics with 25%) because 90% of the filtered Na^+^ load is reabsorbed before reaching the distal convoluted tubule. Furthermore, similar to loop diuretics, thiazides increase K^+^ and H^+^ excretion. Acute effects of thiazides on Ca^2+^ excretion may vary, while chronic administration leads to decreased Ca^2+^ excretion. Thiazide diuretics may increase Mg^2+^ excretion, and long-term use may lead to magnesium deficiency, particularly in the elderly. They are commonly used in combination with loop diuretics in those not satisfactorily treated with loop diuretics alone, and play only a minor role as monotherapy in HF (for review, see [34]).

In the **CLOROTIC** trial [35], patients with AHF were administered either oral hydrochlorothiazide (HCT; 25–100 mg daily, adjusted according to estimated glomerular filtration rate) or a placebo in addition to IV furosemide. Combination diuretic therapy with HCT resulted in a more significant reduction in body weight at 72 h, although it did not yield a statistically significant improvement in patient-reported dyspnea. Notably, HCT was associated with a greater increase in serum creatinine levels. The trial did not reveal any discernible disparities in terms of mortality or rehospitalization rates. The CLOROTIC trial further demonstrated that the addition of HCT led to electrolyte disturbances in up to half of the patients and offers little benefit in patients with renal insufficiency (even if used at higher dosages, up to 100 mg daily). Notwithstanding these limitations, the 2021 ESC guidelines consider thiazides as a class IIa/level of evidence B recommendation for decongestion in AHF patients with resistant edema as a combination therapy with previously administered, adequately dosed IV loop diuretics [36].

### 4.3. Inhibitors of Renal Epithelial Na^+^ Channels (K^+^-Sparing Diuretics)

The only two drugs of this class in clinical use are amiloride and triamterene (Table 4). Both drugs cause small increases in Na^+^ and Cl^−^ excretion and are usually used to offset the hypokalemic effects of other diuretics. Due to this property, amiloride and triamterene (along with mineralocorticoid receptor antagonists) are being referred to as potassium-sparing diuretics. Amiloride and triamterene block epithelial sodium channels (ENaCs) in the luminal membrane of principal cells in late distal tubules and collecting ducts. Due to the limited capacity of the late distal tubule and collecting duct to reabsorb solutes, the Na^+^ and Cl^−^ excretion rate is very modest (~2% of filtered load). Furthermore, these drugs decrease K^+^, H^+^, Ca^2+^, and Mg^2+^ excretion rates. Because of the mild natriuresis induced by these agents, they are usually used only in combination with thiazide or loop diuretics and do not play a relevant role in decongestion in AHF patients.

### 4.4. Mineralocorticoid Receptor Antagonists (K^+^-Sparing Diuretics)

Mineralocorticoids (Table 5) bind to specific mineralocorticoid receptors (MRs) and cause water and salt retention and increased K^+^ and H^+^ excretion. Spironolactone, eplerenone and finerenone competitively inhibit the binding of aldosterone to the MR, thus blocking the biological effects of aldosterone. For this reason, these drugs are also being referred to as aldosterone antagonists. MR antagonists (MRA) are the only diuretics that do not require access to the tubular lumen to induce diuresis. The clinical efficacy of these drugs is dependent on endogenous aldosterone levels, in that higher levels lead to greater effects on urinary excretion. Spironolactone, but not eplerenone, has some affinity toward progesterone and androgen receptors and can thus lead to side effects such as gynecomastia, impotence, and menstrual irregularities. MRA are commonly co-administered with loop diuretics in the treatment of congestion in HF. In AHF patients, high-dose spironolactone (100 mg daily) failed to show incremental benefits for decongestion as compared to its low dose of 25 mg daily, as shown in the randomized double-blind **ATHENA-HF** trial [37]. In 360 AHF patients, high-dose spironolactone did neither change the primary outcome (reduction in log NT-proBNP levels from baseline to 96 h) nor the secondary end points, clinical congestion score, dyspnea assessment, net urine output, or net weight change. The 30-day mortality rate was unaffected as well, and high-dose spironolactone did not induce a higher rate of hyperkalemia or WRF (safety end points).

### 4.5. Carbonic Anhydrase Inhibitors

HF induces a state of increased proximal renal sodium reabsorption (up to 75%), but loop diuretics, thiazides and MRA work in more distal parts of the nephron. Therefore, drugs inhibiting sodium reabsorption in the proximal tubuli such as the carbonic anhydrase inhibitor acetazolamide are increasingly used as the default therapeutic intervention in the decongestion of AHF patients (Table 6). Although acetazolamide is the oldest diuretic still used in clinical practice (being introduced several decades ago), evidence for its efficacy and safety in AHF has just recently been gained by the landmark randomized **ADVOR** trial [38,39].

Acetazolamide is the only carbonic anhydrase inhibitor currently used in the treatment of AHF patients, although other systemic carbonic anhydrase inhibitors such as methazolamide and dichlorphenamide find applications in other medical contexts, for treating certain forms of glaucoma (reduction in intraocular pressure by hindering aqueous humor production via decreasing bicarbonate production) or high-altitude sickness, by counteracting respiratory alkalosis caused by hyperventilation.

Acetazolamide acts within the proximal tubule cells where carbonic anhydrase plays a pivotal role in generating H^+^ and HCO_3_^−^ from water and CO_2_. This process utilizes the Na^+^ gradient to export protons via an Na^+^-H^+^-exchanger (NHE3) into the primary filtrate, simultaneously importing Na^+^ into the tubule cell. HCO_3_^−^ is reclaimed and transported back into the bloodstream via a Na^+^/HCO_3_^−^ co-transporter. Both mechanisms are influenced by angiotensin II. Consequently, serum HCO_3_^−^ levels serve as markers of proximal nephron sodium reabsorption, neurohormonal activation and correlate with diuretic resistance [38]. The speed of these mechanisms largely hinges on carbonic anhydrase activity in tubule cells. Inhibiting carbonic anhydrase reduces the availability of H^+^ for exchange with Na^+^, resulting in a diuretic effect and reduced bicarbonate uptake into the bloodstream, increased bicarbonate loss in urine, counteracting neurohormonal activation and diuretic resistance.

The **ADVOR** trial [39,40] randomized 519 AHF-patients in a 1:1 ratio to IV acetazolamide (500 mg/day for 3 days) or matching placebo on top of standardized IV loop diuretics (dose equivalent of twice oral maintenance dose). The primary endpoint of complete decongestion (morning of day 4) was achieved (42% versus 31%, *p* < 0.001) resulting in a shorter stay in the hospital, but with neutral effects for rehospitalization and/or death (for which the study was underpowered). The treatment response was magnified in patients with baseline or loop-diuretic-induced elevated HCO_3_^−^ by specifically counteracting this component of diuretic resistance. Notably, the study excluded patients undergoing treatment with an SGLT-2 inhibitor, which is now considered a standard therapy for HF (see below).

### 4.6. Sodium-Glucose Co-Transporter 2 (SGLT-2) Inhibitors

Similar to the carbonic anhydrase inhibitor acetazolamide, SGLT-2 inhibitors act on the proximal convoluted tubuli of the nephron (Table 7). However, SGLT-2 inhibitors hardly affect reabsorption of sodium but mainly lead to glucosuria. In people without diabetes mellitus, filtered glucose (about 200 g daily) is virtually fully reabsorbed at the proximal convoluted tubules and no glucose is detected in the urine. Two types of transporters are responsible for the reabsorbed glucose: passive transporters, namely facilitated glucose transporters (GLUTs), as well as active co-transporters, namely sodium-glucose co-transporters. The most important co-transporters, SGLT-1 and SGLT-2, are responsible for reabsorption of virtually 100% of the filtered glucose load.

SGLT-2-inhibitor-induced glucosuria along with the associated osmotic diuresis leads to a reduction in body weight, reduces ventricular preload and myocardial oxygen consumption and decreases the intravascular volume. Furthermore, glucosuria and the accompanied shift in the insulin/glucagon ratio promotes lipolysis in adipose tissue, thereby increasing hepatic ketone production (especially ß-hydroxybutyrate), which are used by the cardiomyocytes as a fuel source for oxidative ATP production. Additionally, SGLT inhibition targets the sodium–hydrogen exchanger (NHE). Two isoforms of NHE exist: NHE 1, which is found in the heart and vasculature, and NHE 3, which is found in the kidneys. NHE 1 inhibition decreases sodium and calcium entry into the cytosol, thus increasing calcium entry into the mitochondria, activating mitochondrial ATP generation and antioxidant enzymes (for review, see [41,42]).

Recent randomized trials have demonstrated the pivotal benefits of SGLT2 inhibitors not only in the treatment of chronic HF (regardless of ejection fraction), but also in AHF. Early initiation of empagliflozin in the **EMPULSE** [43], **EMPA-RESPONSE-AHF** [44,45] and **EMPAG-HF** [46] trials in the standard therapy with loop diuretics for patients hospitalized with AHF is safe and increases urine output without affecting natriuresis. This is probably due to the fact that sodium can be reabsorbed in the entire tubule and a stronger reabsorption of sodium takes place in later sections. Indeed, SGLT2 inhibitors mediate osmotic diuresis mainly through glucosuria rather than natriuresis in patients with AHF. The latter finding may explain why dapagliflozin was not superior to the thiazide-like compound metolazone to receive timely decongestion and weight loss on top of loop diuretics [47].

Patients with HF with reduced ejection fraction shave shown to have abnormally high central and pulmonary venous pressures. These pressures are determined in part by the total blood volume. The part of the total blood volume that does not exert tension on the vessel wall is termed unstressed blood volume, while the volume in excess of the unstressed blood volume is termed the stressed blood volume [48]. Given that the volume status plays a central role in the clinical syndrome of congestive HF, SGLT-2 inhibitors are beneficial in these patients due to the blockage of glucose and associated sodium reabsorption in the proximal renal tubule, thus promoting natriuresis and osmotic diuresis. The **EMPIRE-HF** [49] and **EMPA-RESPONSE-AHF** [44] trials suggest another mechanism of action of SGLT-2 inhibitors, namely sympathoinhibition. Through modulation of the sympathetic nervous system, SGLT-2 inhibitors reduce the stressed blood volume and exercise-induced increases in cardiac filling pressures, which improves exercise tolerance and ameliorates HF symptoms.

At the 2023 ESC congress, results of the **DICTATE-AHF** trial have been presented [50]. The trial investigated the efficacy of early (within 24 h) initiation of the SGLT-2 inhibitor dapagliflozin in patients with AHF. Initially, only patients with type 2 diabetes and an estimated glomerular filtration rate (eGFR) of at least 30 mL/min/1.73 m^2^ admitted to hospital with AHF had been enrolled; however, after a protocol amendment, patients without type 2 diabetes and patients with an eGFR of up to 25 mL/min/1.73 m^2^ could also be enrolled. Type 1 diabetes, systolic blood pressure less than 90 mmHg, serum glucose less than 80 mg/dL, use of IV inotropic therapy, and history of diabetic ketoacidosis constituted the main exclusion criteria. Within 24 h of hospital admission, patients were randomly assigned to oral dapagliflozin 10 mg once daily or usual care until day 5 or hospital discharge. The study did not meet the primary outcome, which was defined as diuretic efficiency (diuretic response) expressed as the cumulative change in weight per cumulative loop diuretic dose (IV and oral). However, dapagliflozin significantly increased both 24 h natriuresis (*p* = 0.025) and 24 h urine output (*p* = 0.005), and decreased both time to completing IV diuretic therapy (*p* = 0.006) and time to hospital discharge (*p* = 0.007). Furthermore, dapagliflozin was shown to be safe across all diabetic and cardiorenal in-hospital outcomes.

### 4.7. Vassopressin-2 Receptor Antagonists

Vasopressin antagonists impede the action of the antidiuretic hormone (ADH) at vasopressin receptor type 2, thereby inhibiting the integration of aquaporins into the walls of the collecting ducts in the kidney (Table 8). This inhibition facilitates the excretion of electrolyte-free water.

The **TACTICS-HF** trial [51] randomized 257 patients in a prospective, double blinded, placebo-controlled trial for treatment with 30 mg of tolvaptan in addition to furosemide given at 0, 24 and 48 h. While the administration of tolvaptan led to increased weight and fluid loss, it was associated with a higher probability of WRF. Additionally, no discernible differences in clinical outcomes were observed. In the smaller **3T Trial** [52], a combined therapy involving tolvaptan and furosemide was assessed in 20 patients. The study revealed a notable enhancement in the efficacy of diuretic therapy, although clinical outcomes were not specifically evaluated in this investigation.

## 5. Non-Pharmacological Interventions in the Prevention and Treatment of AHF

Alternative non-pharmacological concepts such as ultrafiltration or earlier detection of decompensation by device-based remote monitoring need further clarification and their detailed presentation are beyond the scope of this review.

Ultrafiltration therapy represents a technique employed to eliminate excess volume from the vascular system in patients unresponsive to diuretics. This process involves the removal of isotonic plasma water from the bloodstream via a semipermeable membrane, driven by a pressure gradient. Notably, this procedure necessitates the use of a central venous or large lumen peripheral venous catheter as well as anticoagulation.

In contrast to the use of diuretics, ultrafiltration should not cause severe electrolyte disturbances nor hemodynamic instability. In addition, ultrafiltration yields a much better predictable effect as compared to diuretics, and subsequently leads to an improved response to diuretics (for meta-analysis, see [53]).

The benefit of an improved short-term preservation of renal function and a significantly reduced mid-term rehospitalizations rate trials comes at the cost of an increased rate of serious adverse events, shown in randomized trials such as **CARRESS-HF** [54]. ESC guidelines suggest that ultrafiltration may be used in diuretic-resistant patients with volume overload [35], whereas ultrafiltration is not recommended in the American Heart Association (AHA) guidelines. Taken together, ultrafiltration may be an option in selected patients, but constitutes a substantial and cost-intensive intervention with an elevated risk of bleeding.

## 6. Conclusions

Timely and successful decongestion is a main target in AHF, with residual congestion at discharge being associated with worse outcomes. Novel studies have highlighted the potential of combination diuretic therapies, adding acetazolamide, hydrochlorothiazide, or SGLT-2-inhibitors to IV loop diuretics to more effectively achieve euvolemia. Translation into better long-term outcomes, however, still needs to be shown.

## Figures and Tables

**Figure 1 jcm-13-00311-f001:**
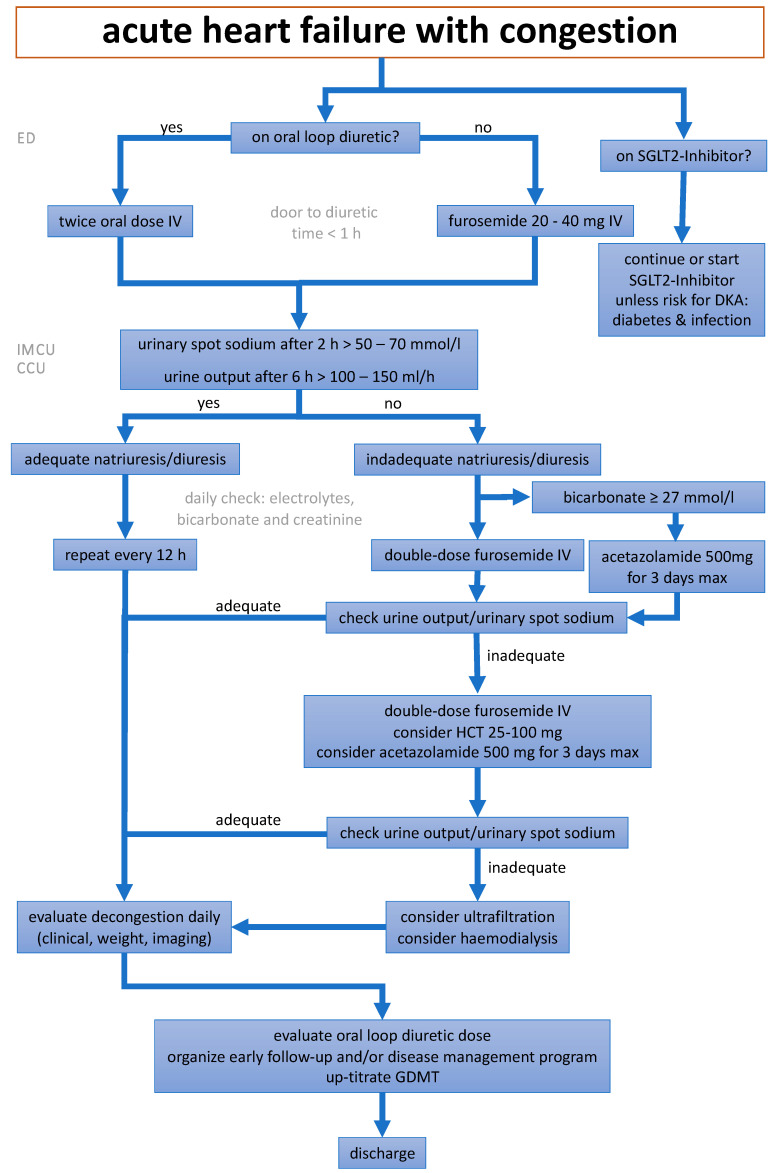
A depiction of the new treatment algorithm for the optimal decongestion strategy, modified from [5]. ED = emergency department, IMCU = intermediate care unit, CCU = coronary care unit, SGLT2 = sodium-glucose co-transporter 2, DKA = diabetic ketoacidosis, HCT = hydrochlorothiazide, GDMT = guideline-directed medical therapy.

**Figure 2 jcm-13-00311-f002:**
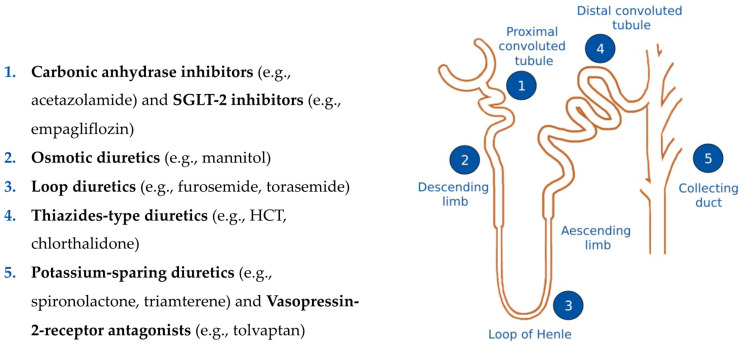
A depiction of the nephron and the different sites of action of the various classes of diuretics. SGLT2 = sodium-glucose co-transporter 2, HCT = hydrochlorothiazide.

**Table 1 jcm-13-00311-t001:** Summary of clinical trials investigating the impact of a natriuresis-guided approach in the first few hours after hospital admission.

Trial Name	Clinical Setting	Trial Description	Treatment	Primary Endpoint	Primary Outcomes
PUSH-AHF [23]	AHF patients requiring treatment with IV loop diuretics	Pragmatic, single-center, randomized, controlled, open-label study	Natriuresis-guided therapy (urine sodium measured at 2, 6, 12, 18, 24 and 36 h) versus usual care	24 h urinary sodium excretion after start of loop diuretic therapy and a combined endpoint of all-cause mortality or first HF rehospitalization at 6 months	The first primary endpoint was met, as natriuresis in the natriuresis-guided and usual care arms was 409 ± 178 mmol arm versus 345 ± 202 mmol, respectively (*p* = 0.0061). However, there were no significant differences between the two arms for the combined endpoint of time to all-cause mortality or first heart failure rehospitalization, which occurred in 46 (31%) and 50 (31%) of patients in the natriuresis-guided and usual care arms, respectively (hazard ratio 0.92, 95% confidence interval 0.62–1.38, *p* = 0.6980).
ENACT-HF [24]	AHF patients on chronic loop diuretic therapy, admitted to the hospital for IV loop diuretic therapy	Prospective, multicenter, open-label, nonrandomized, pragmatic trial	Early assessment of diuretic response with a spot urinary sodium measurement after 2 h and urine output after 6 h. Doubling the initial IV dose after 6 h if urinary sodium < 50 mmol/L or urinary output < 100 mL/h	Total natriuresis after 1 day	Mean natriuresis was 174 mmol in the standard of care arm and 282 in the protocol arm (+64% change; mean ratio: 1.64, 95% confidence interval 1.37–1.95, *p* < 0.001). Hospital duration was 7.0 days in the standard of care group and 5.8 days in the protocol group (mean ratio 0.87, 95% confidence interval: 0.77–0.99, *p* = 0.036).
ESCALATE [25]	Patients with AHF randomized to a diuretic strategy guided by urine chemistry or a usual care strategy	Randomized, double-blind clinical trial	Patients in both arms receive an open-label IV diuretic dose. Patients in the control arm have diuretic dosing based on diuretic response. Patients in the intervention arm have diuretic dosing guided by spot urine chemistry and the natriuretic response prediction equation calculator to achieve the established daily net negative goal.	A composite of the clinical state and global clinical status, assessed daily from randomization to day 14	Trial is still ongoing
DECONGEST	AHF patients with signs of congestion	Pragmatic, multicenter, interventional, parallel-arm, randomized, open-label trial	Diuretic regimen, based on serial assessment of sodium concentration on spot urine samples and with low-threshold use of combination diuretic therapy versus usual care	Mortality, days in hospital and decongestion	Trial is still ongoing

**Table 2 jcm-13-00311-t002:** Inhibitors of Na^+^-K^+^-2Cl^–^ symporter (loop diuretics).

Drug	Relative Potency	Oral Bioavailability (%)	Half-Life (h)	Duration of Action (h)	Typical Oral Doses	Route of Elimination	Side Effects
Furosemide	1	10–90	1–3	6–8	40–200 mg, 1–2 times daily; maximum 600 mg/d	~65% R, ~35% M ^a^	Hypokalemia, hypomagnesemia, hyperuricemia, hypocalcemia, hyponatremia, ototoxicity
Bumetanide	40	80–100	1–3	6–8	0.5–4 mg 1–2 times daily; maximum 10 mg/d	~62% R, ~38% M	
Tora- semide	0.7	80–100	4–6	12–18	20–80 mg/d; maximum 200 mg/d	~20% R, ~80% M	

M, metabolism; R, renal excretion of intact drug. ^a^ Metabolism of furosemide occurs predominantly in the kidney.

**Table 3 jcm-13-00311-t003:** Inhibitors of Na^+^-Cl^−^ symport (thiazide diuretics).

Drug	Relative Potency	Oral Bioavailability (%)	Half-Life (h)	Duration of Action (h)	Typical Oral Doses	Route of Elimination	Side Effects
**Thiazide diuretics**							Hypokalemia, hypomagnesemia, hypercalcemia, hyponatremia, hyperuricemia, sulfonamide allergy
Hydrochloro thiazide	1	~70	~2.5	6–12	12.5–100 mg/d Maximum: 200 mg/d	R	
Chlorothiazide (IV formulation available)	0.1	9–56 (dose-dependent)	~1.5	6–12	500–1000 mg/d Maximum: 1000 mg/d	R	
**Thiazide-like diuretics**							
Metolazone	10	~65	8–14	≥24	2.5–10 mg/d Maximum: 20 mg/d	~80% R, ~10% B, ~10% M	
Chlorthalidone	1	~65	~47	24–72	12.5–25 mg/d Maximum: 100 mg/d	~65% R, ~10% B, ~25% U	
Indapamide	20	~93	~14	≥24	2.5 mg/d Maximum: 5 mg/d	M	

B, excretion of intact drug into bile; M, metabolism; R, renal excretion of intact drug; U, unknown pathway of elimination.

**Table 4 jcm-13-00311-t004:** Inhibitors of renal epithelial Na^+^ channels (K^+^-sparing diuretics).

Drug	Relative Potency	Oral Bioavailability (%)	Half-Life (h)	Duration of Action (h)	Typical Oral Doses	Route of Elimination	Side Effects
Amiloride	1	15–25	~21	~24	5–10 mg/d Maximum: 20 mg/d	R	Hyperkalemia
Triamterene	0.1	~50	~4	7–9	100 mg twice daily Maximum: 300 mg/d	M	

M, metabolism; R, renal excretion of intact drug.

**Table 5 jcm-13-00311-t005:** Mineralocorticoid receptor antagonists (K^+^-sparing diuretics).

Drug	Oral Bioavailability (%)	Half-Life (h)	Duration of Action (h)	Typical Oral Doses	Route of Elimination	Side Effects
Spironolactone	~65	~1.6	48–72	25–50 mg/d Maximum: 200 mg/d	M	Hyperkalemia Spironolactone: gynecomastia
Eplerenone	69	~5	~48	25 mg/d Maximum: 50 mg/d	M	
Finerenone	44	2–3	~48	20 mg/d Maximum: 20 mg/d	M	

M, metabolism.

**Table 6 jcm-13-00311-t006:** Carbonic anhydrase inhibitor.

Drug	Oral Bioavailability (%)	Half-Life (h)	Duration of Action (h)	Typical Oral Doses	Route of Elimination	Side Effects
Acetazolamide	70–90	3–9	8–12	250–500 mg/day	R	Metabolic acidosis Hypokalemia Hyponatremia Hyperchloremia Alkalinization of urine

R, renal excretion of intact drug.

**Table 7 jcm-13-00311-t007:** Sodium-glucose co-transporter-2 (SGLT-2) inhibitors.

Drug	Oral Bioavailability (%)	Half-Life (h)	Duration of Action (h)	Typical Oral Doses	Route of Elimination	Dose Modifications
Empagliflozin	78	12.4	~72	10 mg/d Maximum: 25 mg/d	~ 50% M, ~ 50% R	eGFR ≥ 45 mL/min/1.73 m^2^: No dosage adjustment required eGFR 30–45 mL/min/1.73 m^2^: Do not initiate therapy; if already on it, discontinue therapy when eGFR persistently <45 mL/min/1.73 m^2^ eGFR < 30 mL/min/1.73 m^2^: Contraindicated
Dapagliflozin	78	~12.9	~72	5 mg/d Maximum: 10 mg/d	21% M, 79% R	eGFR ≥ 45 mL/min/1.73 m^2^: No dosage adjustment required eGFR 30 to <45 mL/min/1.73 m^2^: Not recommended eGFR < 30 mL/min/1.73 m^2^: Contraindicated
Canagliflozin	65	10–13 (dose-dependent)	~24	100 mg/d Maximum: 300 mg/d	41.5% M, 7% as hydroxylated metabolite, 3.2% as O-glucuronide metabolite, <1% R, 30.5% as O-glucuronide metabolites	eGFR ≥ 60 mL/min/1.73 m^2^: No dosage adjustment necessary eGFR 45 to <60 mL/min/1.73 m^2^: 100 mg qDay eGFR 45 to <60 mL/min/1.73 m^2^ with albuminuria > 300 mg/day: 100 mg qDay eGFR < 30 mL/min/1.73 m^2^ or end-stage kidney disease on dialysis: Contraindicated

M, metabolism; R, renal excretion of intact drug.

**Table 8 jcm-13-00311-t008:** Vasopressin receptor antagonists.

Drug	Oral Bioavailability (%)	Half-Life (h)	Duration of Action (h)	Typical Oral Doses	Route of Elimination	Side Effects
Tolvaptan	56 (42–80)	3 (15 mg)–12 (≥120 mg)	~24	15–60 mg/d Maximum: 60 mg/d	M	Thirst, dry mouth, pollakiuria or polyuria, asthenia, constipation, hyperglycemia, pyrexia, anorexia
Conivaptan	NA	5	~24	IV route only; 20 mg as a loading dose over 30 min, followed by continuous infusion of 20 mg over 24 h for 2–4 days. Maximum: 40 mg/d	M	Infusion site reactions (e.g., erythema, pain, phlebitis), hypokalemia, headache, peripheral edema, vomiting, diarrhea, constipation, hypertension, orthostatic hypotension, hyponatremia, thirst, anemia, hypotension, pyrexia, nausea, confusion

M, metabolism.

## Data Availability

Not applicable.

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
