# Peer review of "Decongestion in Acute Heart Failure—Time to Rethink and Standardize Current Clinical Practice?"

_jcm, 2024, doi:10.3390/jcm13020311_

Round 1

Reviewer 1 Report

Comments and Suggestions for Authors

Extremely well written review with balanced view.

My only suggestion is that since results of DICTATE-AHF are  available , these should be incorporated. Especially, natriuresis has been shown to be significant with dapagliflozin and also clinical outcome with regard to  early discharge

Some discussion about stressed blood volume in relationship with EMPIRE and EMPA-Response-AHF will be in order. This line of thinking might delve upon another way of benefit  by SGLT2 inhibitors in AHF

Author Response

Dear reviewer, 
Thank you for your critical, quick and encouraging review. 
We have revised our manuscript accordingly. Please find our point-by-point response below: 
Extremely well written review with balanced view.
My only suggestion is that since results of DICTATE-AHF are available, these should be incorporated. Especially, natriuresis has been shown to be significant with dapagliflozin and also clinical outcome with regard to early discharge.
Some discussion about stressed blood volume in relationship with EMPIRE and EMPA-Response-AHF will be in order. This line of thinking might delve upon another way of benefit by SGLT2 inhibitors in AHF.
We thank the reviewer for the valuable and insightful comments. We have included the results of the DICTATE-AHF trial and elaborated on the impact of SGLT-2 inhibitors on the stressed blood volume through sympathoinhibition and the resulting improvement in HF symptoms. The elaborated discussion of these two topics can be found on page 10. 

Reviewer 2 Report

Comments and Suggestions for Authors

I had the pleasure of reviewing the article entitled Decongestion in Acute Heart Failure – Time to Rethink and 2 Standardize Current Clinical Practice?”

The paper provides an excellent overview of a very important clinical issue related to diuretic treatment in patients with acute/decompensated heart failure.

The article is very well written, systematically organizing in a very useful manner information related to the physiopathology of volume retention in heart failure and its treatment using various classes of medications.

In the paper were also discussed the main clinical studies that provide information intended to guide the approach of decongestion and prognostic parameters necessary for evaluating the response to diuretic therapy.

I appreciate the article as being particularly useful for clinicians, but I have one minor suggestion: please summarize in addition the studies describe in subsection 2 “natriuresis guided approach in the first hours “in a table.

Author Response

Dear reviewer, 
Thank you for your critical, quick and encouraging review. 
We have revised our manuscript accordingly. Please find our point-by-point response below: 
I had the pleasure of reviewing the article entitled „Decongestion in Acute Heart Failure – Time to Rethink and Standardize Current Clinical Practice?”
The paper provides an excellent overview of a very important clinical issue related to diuretic treatment in patients with acute/decompensated heart failure.
The article is very well written, systematically organizing in a very useful manner information related to the physiopathology of volume retention in heart failure and its treatment using various classes of medications.
In the paper were also discussed the main clinical studies that provide information intended to guide the approach of decongestion and prognostic parameters necessary for evaluating the response to diuretic therapy.
I appreciate the article as being particularly useful for clinicians, but I have one minor suggestion: please summarize in addition the studies describe in subsection 2 “natriuresis guided approach in the first hours “in a table.
We thank the reviewer for the valuable and insightful comments. We have included a table (Table 1) on pages 4-5 which summarizes the clinical trials investigating the impact of a natriuresis guided approach in the first hours after hospital admission. 

Reviewer 3 Report

Comments and Suggestions for Authors

The authors provide an interesting review on diuretic treatment in heart failure. However, I find it incomplete as, in addition to the review, it lacks an integrative perspective that offers a solution to the question posed in the title. I see several changes that could enhance the article:

Major Changes:

  • A section preceding the conclusions is missing, which should propose (along with a scheme, figure, or table) a treatment algorithm and objectives for natriuresis, weight loss, etc., in the short and medium term with a specific diuretic or a combination thereof based on the clinical scenario. This is complex but necessary as it is the only way to advance knowledge with this article.

Minor Changes:

  • In the abstract and text, it is specified that the ceiling dose is 200 mg of furosemide; however, much higher doses are used in clinical practice. I would remove this figure as I believe it does not contribute.
  • Page 3, Lines 165-171: This paragraph discusses the "ENACT-HF" study but does not state its conclusions, which I find very interesting and complementary to the information provided. They should be included.
  • Page 4, Paragraph starting at Line 225: It is specified that intravenous loop diuretics should be discontinued after the third day. The next sentence mentions that these drugs should be withdrawn if possible. On what evidence are these statements based? It should be cited.
  • On page 9, in the paragraph about SGLT2 inhibitors, it is mentioned that despite being inhibitors of the sodium-glucose cotransporter, they do not increase natriuresis (EMPULSE and EMPA-RESPONSE studies). It would be appropriate to explain in a couple of sentences how the authors interpret this issue.

Author Response

Dear reviewer, 
Thank you for your critical, quick and encouraging review. 
We have revised our manuscript accordingly. Please find our point-by-point response below: 
The authors provide an interesting review on diuretic treatment in heart failure. However, I find it incomplete as, in addition to the review, it lacks an integrative perspective that offers a solution to the question posed in the title. I see several changes that could enhance the article:
Major Changes:
A section preceding the conclusions is missing, which should propose (along with a scheme, figure, or table) a treatment algorithm and objectives for natriuresis, weight loss, etc., in the short and medium term with a specific diuretic or a combination thereof based on the clinical scenario. This is complex but necessary as it is the only way to advance knowledge with this article.
This is an interesting point. Our new treatment algorithm is now presented in figure 1. 

Minor Changes:
In the abstract and text, it is specified that the ceiling dose is 200 mg of furosemide; however, much higher doses are used in clinical practice. I would remove this figure as I believe it does not contribute.
Thank you for this insightful comment. We have revised the doses for furosemide according to the current ESC guideline recommendations.
Page 3, Lines 165-171: This paragraph discusses the "ENACT-HF" study but does not state its conclusions, which I find very interesting and complementary to the information provided. They should be included.
Thank you for this valuable comment. We have included the conclusions of the ENACT-HF trial on page 3.
Page 4, Paragraph starting at Line 225: It is specified that intravenous loop diuretics should be discontinued after the third day. The next sentence mentions that these drugs should be withdrawn if possible. On what evidence are these statements based? It should be cited.
Thank you for this comment. We added some citations, although switching to oral therapy is only based on expert opinion.
On page 9, in the paragraph about SGLT2 inhibitors, it is mentioned that despite being inhibitors of the sodium-glucose cotransporter, they do not increase natriuresis (EMPULSE and EMPA-RESPONSE studies). It would be appropriate to explain in a couple of sentences how the authors interpret this issue.
Thank you for this comment. We added an interpretation of these results, although other studies like DICTATE-AHF suggest other findings.

Round 2

Reviewer 3 Report

Comments and Suggestions for Authors

The authors have systematically addressed and implemented the suggested changes, point by point. They have even added a new therapeutic algorithm, which is, at the very least, intriguing. The article is now more readable and engaging. From my perspective, I have no additional comments; it is a highly interesting review, and I believe it is straightforward to read.

Author Response

We would like to thank the reviewer for his extremely valuable input and kind words. We also believe that the algorithm has substantially improved our manuscript. We really appreciate his suggestions raised in the first round of the review.